# Maternal mortality in Malawi, 1977–2012

Tim Colbourn,[1] Sonia Lewycka,[1] Bejoy Nambiar,[1] Iqbal Anwar,[2] Ann Phoya,[3] Chisale Mhango[4]

## ABSTRACT

**Background:** Millennium Development Goal 5 (MDG 5) targets a 75% reduction in maternal mortality from 1990 to 2015, yet accurate information on trends in maternal mortality and what drives them is sparse. We aimed to fill this gap for Malawi, a country in sub-Saharan Africa with high maternal mortality.

**Methods:** We reviewed the literature for population-based studies that provide estimates of the maternal mortality ratio (MMR) in Malawi, and for studies that list and justify variables potentially associated with trends in MMR. We used all population-based estimates of MMR representative of the whole of Malawi to construct a best-fit trend-line for the range of years with available data, calculated the proportion attributable to HIV and qualitatively analysed trends and evidence related to other covariates to logically assess likely candidate drivers of the observed trend in MMR.

**Results:** 14 suitable estimates of MMR were found, covering the years 1977–2010. The resulting best-fit line predicted MMR in Malawi to have increased from 317 maternal deaths/100 000 live-births in 1980 to 748 in 1990, before peaking at 971 in 1999, and falling to 846 in 2005 and 484 in 2010. Concurrent deteriorations and improvements in HIV and health system investment and provisions are the most plausible explanations for the trend. Female literacy and education, family planning and poverty reduction could play more of a role if thresholds are passed in the coming years.

**Conclusions:** The decrease in MMR in Malawi is encouraging as it appears that recent efforts to control HIV and improve the health system are bearing fruit. Sustained efforts to prevent and treat maternal complications are required if Malawi is to attain the MDG 5 target and save the lives of more of its mothers in years to come.

[1]UCL Institute for Global Health, London, UK
[2]International Centre for Diarrhoeal Diseases Research, Dhaka, Bangladesh
[3]Government of the Republic of Malawi, Ministry of Health Sector-Wide Approach (SWAp), Lilongwe, Malawi
[4]Ministry of Health Reproductive Health Unit, Government of the Republic of Malawi, Lilongwe, Malawi

**Correspondence to**
Dr Tim Colbourn;
t.colbourn@ucl.ac.uk

### Strengths and limitations of this study

- The study provides the most detailed review of trends in maternal mortality in Malawi to date, including the estimation of trends in the maternal mortality ratio, comparison with WHO and Institute for Health Metrics and Evaluation estimates and assessment of the variables most likely to have driven the trend.
- It includes quantitative estimation of the impact of HIV and antiretroviral treatment on maternal mortality in Malawi.
- Sparse data precluded the possibility of quantitatively modelling the relationships between potential explanatory variables and maternal mortality in Malawi.
- The study is comprehensive and conducted by researchers with extensive knowledge and experience of maternal health in Malawi; however, it is not a systematic review.

mortality from population-based studies in Malawi, and explore trends and possible reasons behind them, in order to gauge progress towards achieving the MDG.

The maternal mortality ratio (MMR) is the most common measure of maternal mortality and is expressed as the number of maternal deaths/100 000 live-births, where a maternal death is defined as "the death of a woman while pregnant or within 42 days of the termination of pregnancy, irrespective of the duration and the site of the pregnancy, from any cause related to or aggravated by the pregnancy or its management but not from accidental or incidental causes."[4] Maternal deaths can be divided into direct and indirect obstetric deaths. Direct obstetric deaths are defined as "those resulting from obstetric complications of the pregnant state (pregnancy, labour and puerperium), from interventions, omissions, incorrect treatment or from a chain of events resulting from any of the above"[4] and indirect obstetric deaths defined as "those resulting from previous existing disease or disease that developed during pregnancy and which was not due to

## BACKGROUND

Maternal mortality in Malawi is high; the most recent national survey estimate is 675 maternal deaths/100 000 live-births during the period 2004–2010.[1] Millennium Development Goal (MDG) 5 aims to reduce maternal mortality by 75% between 1990 and 2015.[2] This equates to a reduction from 620 maternal deaths/100 000 live-births in 1990[3] to 155 by 2015. We review data on maternal

direct obstetric causes, but which was aggravated by physiological effects of pregnancy."[4] Data pertaining to obstetric deaths are not always available. Deaths occurring during pregnancy, childbirth and puerperium, hereafter referred to as 'pregnancy-related' deaths are defined as "death occurring during pregnancy, childbirth and puerperium is the death of a woman while pregnant or within 42 days of termination of pregnancy, irrespective of the cause of death (obstetric and non-obstetric)."[4] Given a lack of adequate information on causes of death, surveys reporting maternal mortality often rely only on the timing of the death in relation to pregnancy and therefore report pregnancy-related mortality as MMR. In settings such as Malawi, where it is difficult to distinguish between HIV-disease-related indirect obstetric deaths and incidental deaths due to HIV coincident with pregnancy, this is also more likely.

The leading biological causes of maternal death in Africa are haemorrhage, infections and hypertensive disorders,[5] and these deaths are mediated by a complex set of underlying social, economic and behavioural factors, typically grouped into the 'Three Delays'.[6] The delay by the patient in the decision to seek care, the delay in reaching the appropriate care once the decision has been made to seek care and the delay in receiving adequate care after arriving at the health facility, all contribute to maternal mortality. Dynamics in the drivers of these delays and in interventions to ameliorate them and treat the biological causes of maternal death they allow, all contribute to changing trends in maternal mortality.[7]

## METHODS
### Review of MMR data in Malawi
We searched for studies concerning maternal mortality in Malawi, primarily via PubMed and Google Scholar, but also including Demographic and Health Survey (DHS) reports and those from the United Nations (UN) and WHO. The search term: "(maternal OR pregnancy-related) AND (mortality OR death) AND Malawi" retrieved 203 articles on PubMed in a final search on 28 October 2013. Abstracts were screened and full texts retrieved when the abstract indicated data on population-level maternal mortality—that is, purely facility-based studies were excluded. This search yielded four studies, all reporting subnational population-based estimates of MMR. All national-level estimates of MMR were obtained from DHS and Multiple Indicator Cluster Survey (MICS) reports; our combined total of over 70 (concurrent) years of experience working in maternal health in Malawi has not made us aware of any additional studies. The following information was extracted: date, location and method of survey, case definition of maternal death, number of maternal deaths and livebirths and MMR and CIs (table 1). Studies containing all of this information, and definitions of maternal death or pregnancy-related death analogous to those

stated in the introduction were considered to be of adequate quality for inclusion in our review. No identified studies were rejected on quality grounds and there were no disagreements among authors on which studies to include, or on the data extracted.

Two of the data sources for maternal mortality estimates are from prospective population-based surveillance systems in the central region of Malawi. In both systems, community-based enumerators collected information about pregnancies, births and deaths, and deaths were followed up by verbal autopsies conducted by field supervisors. In MaiMwana project, in Mchinji district, the enumerators were paid staff who reported to field interviewers and supervisors, who followed up with further postpartum interviews.[8] Data were included for control areas of the study (with no interventions), for the period 1 January 2005–31 January 2009.[9] In MaiKhanda, in Lilongwe, Salima and Kasungu districts, the enumerators were volunteers who covered a smaller population each, and who reported to community health workers, officially referred to as health surveillance assistants. Given the lack of an observed effect of the MaiKhanda interventions on maternal mortality, data were included for all areas for the period between 1 July 2007–31 December 2010.[10]

Not all of the published studies reported 95% CIs for their estimates, so these were calculated using the Newcombe-Wilson method without continuity correction.[11] Data from the 1992 DHS were reanalysed in a study by ORC Macro.[12]

We estimate the trend in the MMR in Malawi from 1977 to 2010 using the best-fit multi-term fractional polynomial transformation based on the available population-level data representing the whole country, using the *fracpoly* command in Stata 13.0 for Mac. Further details are provided in web appendix 1. We compare this trend to that estimated by recent modelling studies.

### Review of drivers of MMR in Malawi
A list of possible variables that could have impacted on MMR in Malawi was drawn up using results from modelling studies (ref. 13, p.85][14–16] and relevant overviews.[7 17 18] Literature and Internet-based databases were used to obtain data on the levels of each of the relevant variables in Malawi over the past 35 years. Identified variables were: percentage of deliveries attended by skilled personnel, percentage of deliveries by caesarean section (C-section), total fertility rate (TFR), gross national income (GNI) per capita, health expenditure per capita, life expectancy at birth, female illiteracy rate, HIV prevalence and access to antiretroviral therapy (ART), political stability, malaria, malnutrition and variables associated with the accuracy of MMR data collection. The trends in these variables and in intermediate variables concerned with their mechanisms of action were then compared to the trends in MMR in order to qualitatively and logically assess whether they might have contributed to changes in MMR in Malawi.

**Table 1** Population-based studies and analyses of maternal mortality in Malawi

| Study | Year | Method | Case definition | Location | Maternal deaths | MMR (95% CI)* |
|---|---|---|---|---|---|---|
| Chiphangwi et al[1] | 1977–1979 | Indirect sisterhood† | Deaths of sisters who died "during pregnancy, childbirth or within 6 weeks of giving birth" | Southern region (Thyolo district) | 150 | 409 (349 to 480) |
| Malawi DHS 1992 (reanalysis)[22] | 1977–1983 | Direct sisterhood | Deaths during pregnancy, childbirth and up to 6 weeks afterwards | Malawi (all regions) | | 269 (??) |
| Malawi DHS 1992 (reanalysis)[22] | 1979–1985 | Direct sisterhood | Deaths during pregnancy, childbirth and up to 6 weeks afterwards | Malawi (all regions) | 42 | 408 (242 to 575) |
| McDermot et al[2] | 1987–1989 | Prospective cohort | Deaths during pregnancy, childbirth and up to 6 weeks afterwards | Southern region (Mangochi district) | 15 | 398 (241 to 656) |
| Malawi DHS 1992 ‡[3] | 1986–1992 | Direct sisterhood | Deaths during pregnancy, childbirth and up to 6 weeks afterwards | Malawi (all regions) | 71 | 620§ (410 to 830) |
| Malawi DHS 1992 (reanalysis)[22] | 1986–1992 | Direct sisterhood | Deaths during pregnancy, childbirth and up to 6 weeks afterwards | Malawi (all regions) | 82 | 752 (497 to 1006) |
| Beltman et al[3] | 1994–1996 | Indirect sisterhood† | Deaths of sisters who died "during pregnancy, childbirth or within 6 weeks of giving birth" | Southern region (Thyolo district) | 84 | 558 (260 to 820) |
| Malawi DHS 2000‡[75] | 1994–2000 | Direct sisterhood | Deaths during pregnancy, childbirth and up to 2 months afterwards | Malawi (all regions) | 344 | 1120 (950 to 1288) |
| Malawi DHS 2004‡[45] | 1998–2004 | Direct sisterhood | Deaths during pregnancy, childbirth and up to 2 months afterwards | Malawi (all regions) | 240 | 984 (804 to 1164) |
| MICS 2006‡[23] | 2000–2006 | Direct sisterhood | Deaths during pregnancy, childbirth and up to 2 months afterwards | Northern region | 33 | 543 (325 to 761) |
| | | | | Central region | 190 | 678 (529 to 828) |
| | | | | Southern region | 246 | 1029 (840 to 1217) |
| | | | | Urban Malawi | 77 | 861 (492 to 1230) |
| | | | | Rural Malawi | 392 | 802 (689 to 915) |
| | | | | Malawi (all regions) | 469 | 807 (696 to 918) |
| van den Broek et al[4] | 2002 | Household survey | "The death of a woman associated with childbirth". Timing not stated | Southern region (rural) | 9 | 413 (144 to 682) |
| Malawi DHS 2010‡[1] | 2004–2010 | Direct sisterhood | Deaths during pregnancy, childbirth and up to 2 months afterwards | Malawi (all regions) | 331 | 675 (570 to 780) |
| MaiMwana (control arm)[9] | 2006–2009 | Surveillance (prospective) | WHO ICD10 maternal death (see Background section) All 29 maternal deaths were verified by verbal autopsy | Central region (Mchinji district, rural) | 29 | 585 (407 to 838) |
| MaiKhanda (total¶)[10 25] | 2007–2010 | Surveillance (prospective) | WHO ICD10 maternal death (see Background section), 51/102 (50%) verified by verbal autopsy, the rest verified by call-backs to community | Central region (Kasungu, Lilongwe and Salima districts, rural) | 102 | 299 (247 to 363) |

*Calculated as (100 000/MMR)×maternal deaths; for sisterhood studies calculated as ((maternal deaths/exposure years)/general fertility rate)×100 000.
†Should not be compared with direct sisterhood method as although the reference period is on average 12 years before the survey it includes recent deaths, which will bias the 12-year-old estimate upwards given that the MMR in Malawi increased in the 1990s.
‡The lower 95% CI of the MMR is calculated from the upper 95% CI of the GFR (less deaths/more births) and visa versa. Fertility is only reported for the whole sample in the MICS survey, therefore only the whole sample MMR (Malawi (all regions)) could be recalculated.
§In the 1992 DHS, the GFR used to calculate MMR is stated as 0.220 (table 11.4, p. 123). However in chapter 3 on fertility the total GFR is stated as 223/1000 women (or 0.223; table 3.1, p. 19) but this is for women aged 15–44 only. Using the raw data on number of women interviewed weighted by population of each cluster (district) so that the sample is representative of the whole of Malawi (table 2.8.2, p. 15) results in a total GFR/woman aged 15–49 years of 0.208 which is different to the 0.220 used to arrive at the MMR of 620 produced in the report.
¶Given the MaiKhanda trial showed no effect of either of the interventions on maternal mortality.
DHS, Demographic and Health Survey; GFR, general fertility rate; MICS, Multiple Indicator Cluster Survey; MMR, maternal mortality ratio.

Bivariate and multivariate linear regressions were run, but a lack of data points reduced the power of these models to detect significant associations, especially non-linear associations and interaction terms—both of which are likely given the complex nature of potential causal pathways between the variables and MMR, therefore these results are not included. The lack of data points also precluded the use of multiple imputation to estimate MMR and the effects of the potential predictor variables for the years with missing data.

We used the relative risk (RR) of pregnancy-related mortality in HIV-positive compared to HIV-negative women estimated by Calvert and Ronsmans via systematic review and meta-analysis to be 7.74,[19] to estimate the proportion of the MMR that is due to HIV depending on the HIV prevalence. We adjusted this proportion downwards from 2003 to account for the increasing impact of the ART programme in Malawi.[20] [21] Further details are provided in web appendix 2.

## RESULTS
### Studies of population MMR in Malawi
Eleven studies reporting population-based estimates of MMR were found, and one study contributed four separate estimates[3] [22] (table 1). All DHS and MICS studies,

that is, all of the studies providing national-level estimates, use pregnancy-related deaths as case-definitions of maternal deaths to calculate MMR.

### Trend in national population-based estimates of MMR
Figure 1 plots the MMR of the eight analyses, each applying to a presurvey time period of a number of years, from the five surveys representative of the whole of Malawi (the four DHS surveys and the MICS survey) and the other surveys representative of specific regions of Malawi (table 1). It appears that from a low of around 400 in the early 1980s the MMR rose rapidly throughout the 1980s and early to mid-1990s to a peak of over 1000 maternal deaths/100 000 live-births around 1997. The MMR then fell to around 800 by 2003 and to an estimated 675 in 2007.[1]

Figure 1 plots locally weighted (Lowess) regression trends for the estimates and 95% CI of the national-level surveys in black. The best-fit fractional polynomial transformation is plotted in gold, details of its calculation are provided in web appendix 1. A comparison of this best-fit trend line and recent modelling studies is provided in table 2. The WHO estimates of MMR for Malawi, developed by the Maternal Mortality Estimation Inter-Agency Group (MMEIG)[15] are based on a model

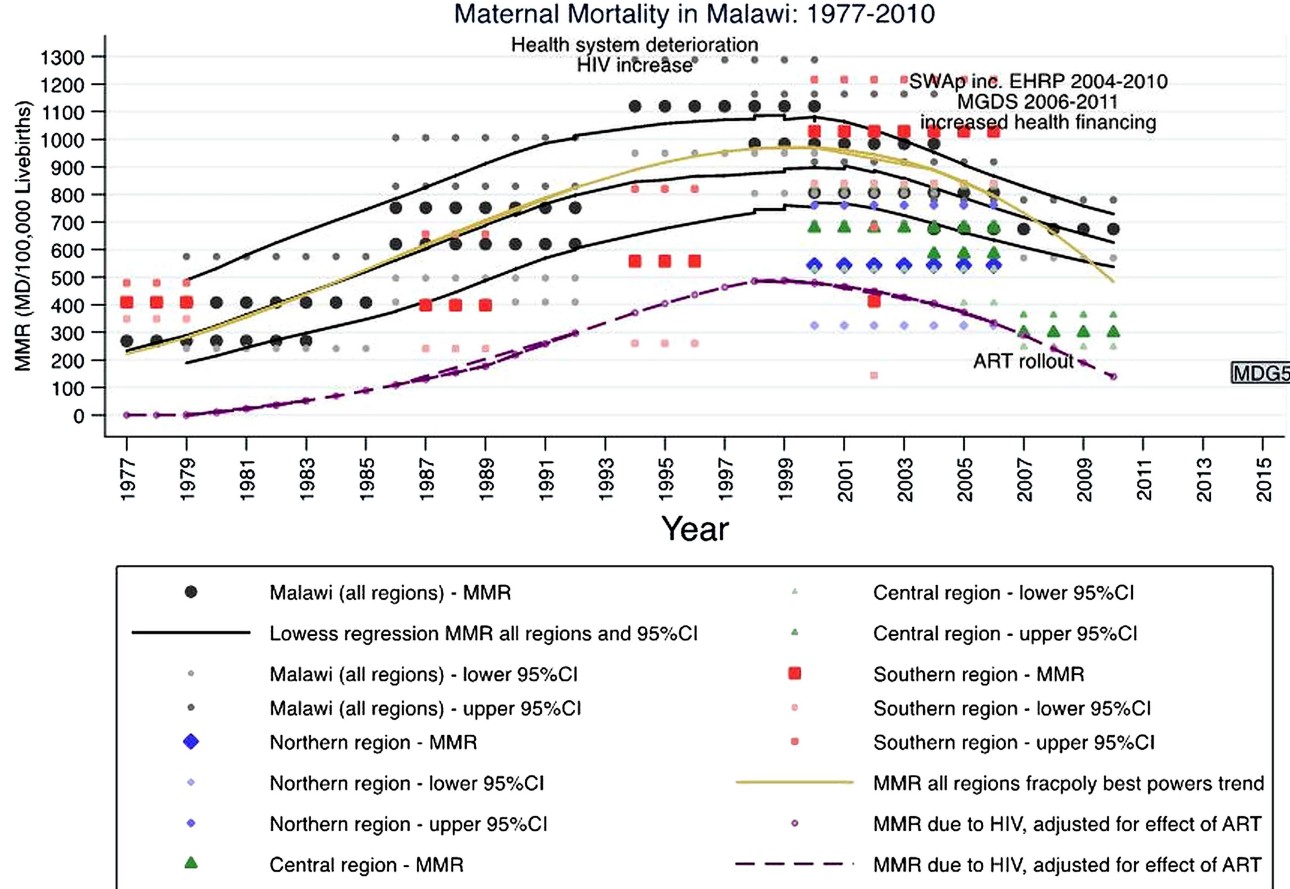

**Figure 1** Trends in maternal mortality in Malawi and its southern, central and northern regions, and estimated maternal mortality due to HIV, 1977–2010.

**Table 2** Comparison of estimated trends in the MMR in Malawi from 1990 to 2010

| 1977 | 1990 | 1995 | 2000 | 2005 | 2010 | Source |
|------|------|------|------|------|------|--------|
| 223 | 748 | 916 | 970 | 846 | 484 | This paper, all years of survey* |
| | 606 | | 1397 | | 422 | IHME |
| | 1100 | 1000 | 840 | 630 | 460 | MMEIG |

*Best-fitting fractional polynomial transformation of MMR by year, using estimates for all years covered by each survey, see web appendix 1 for explanation.
IHME, Institute for Health Metrics and Evaluation; MMEIG, Maternal Mortality Estimation Inter-Agency Group; MMR, maternal mortality ratio.

fitted for all countries of the world and therefore may not accurately convey country-specific nuances. This seems very likely for Malawi as the higher 1990 figure and steady decline (table 2) seem to ignore the evidence suggesting a rise in the 1990s followed by a fall in the 2000s (figure 1). Despite also being generalised to many countries, a different model developed by the Institute for Health Metrics and Evaluation (IHME) does capture this rise and fall,[16] although they estimate a much higher peak around the year 2000 than our estimate (table 2). The MMEIG and IHME estimates differ as they are obtained from statistical models containing different covariates. The MMEIG uses gross domestic product per capita, general fertility rate and skilled birth attendance (SBA).[15] IHME not only uses analogous measures, but also uses antenatal care coverage, female education by age, ART-adjusted HIV prevalence, neonatal death rate and malnutrition.[16] In addition, it is unclear whether the IHME estimates are based on exactly the same national-level MMR data sources as the MMEIG estimates (which use DHS and MICS), because the sources of the expanded set of sibling history data used by IHME are not disclosed in their paper or web appendix.[16] Our best-fitting estimate only uses national-level MMR data by year without using any additional covariates.

### Trend in estimates of MMR in central region of Malawi
Data from prospective surveillance of populations in Mchinji (MaiMwana) and Salima, Lilongwe and Kasungu (MaiKhanda) give MMRs of 585 for MaiMwana during 2005–2008 and 299 for MaiKhanda during 2007–2010 (table 1 and figure 1). These numbers, which are based on obstetric deaths, are lower than the most recent national estimates from DHS and MICS reports, which are based on pregnancy-related deaths (table 1). This may, however, also reflect lower MMR in the central region of Malawi, which was estimated to be 678, compared to the national estimate of 807 in the MICS report, during 2000–2006 (table 1). Although this difference is not statistically significant (the 95% CIs overlap), the 2006 MICS reports the central and northern regions of Malawi to have significantly lower MMR than the

southern region[23] (table 1). Comparison of DHS data by region was not possible because the place of exposure and death of the respondents' sisters who died is not recorded.[24]

### Variables linked to changes in MMR
Figure 2 conceptualises potential drivers of maternal mortality in Malawi. These linkages, and the evidence supporting them, are described below. Table 3 accompanies this analysis by examining how trends in each of the explanatory variables relate to the estimated trend of maternal mortality.

#### Skilled birth attendance, C-section and emergency obstetric care
The proportion of women attended by a skilled health professional at delivery has changed very little over the majority of the period in question, so is unlikely to have contributed to the observed changes in MMR. Similarly, although home deliveries attended by relatives or friends have decreased, they appear to have shifted to deliveries by traditional birth attendants resulting in a similar proportion of deliveries remaining unskilled during all but the most recent years of the period in question (table 3). The DHS 2010 results[1] showed that SBA increased to 71% in the 6 years to 2010, however, while MMR fell to 675. This is encouraging, however further declines in maternal mortality could perhaps be possible if it were not for the overcrowding of facilities and the fact that human and material resources for health have not kept pace with the recent rapid rise in women delivering at facilities.[25]

The lack of association of SBA with MMR questions the validity of the indicator 'delivery by a skilled birth attendant' as a proxy for MMR, especially on consideration that many Asian countries have achieved a lower MMR with much lower skilled attendance at birth, for example, Bangladesh[26 27] and Nepal.[28 29] It is also possible that both the actual level of skills and knowledge of the attendants and the ability of the surrounding environment (eg, drugs and supplies) to ensure the possibility of truly skilled attendance including provision of basic and comprehensive emergency obstetric care has altered significantly over the period in question.[30] However, there are insufficient data to determine whether such changes took place in Malawi. In addition, there has been no adequate verification of the reporting of attendance by nurses, midwives and doctors (taken by the DHS to be 'skilled attendance') by the women surveyed to arrive at these statistics.[30 31]

Remaining below the WHO recommended minimum level of 5%, the C-section rate in Malawi has been consistently low throughout the past 20 years (table 3). It is therefore unlikely to be associated with the observed trends in MMR. It is also important to note that we are unaware of the indications of these C-sections and in particular whether they were undertaken for life-saving purposes. When the population-based C-section rate is

**Figure 2** Overview of variables linked to maternal mortality in Malawi.

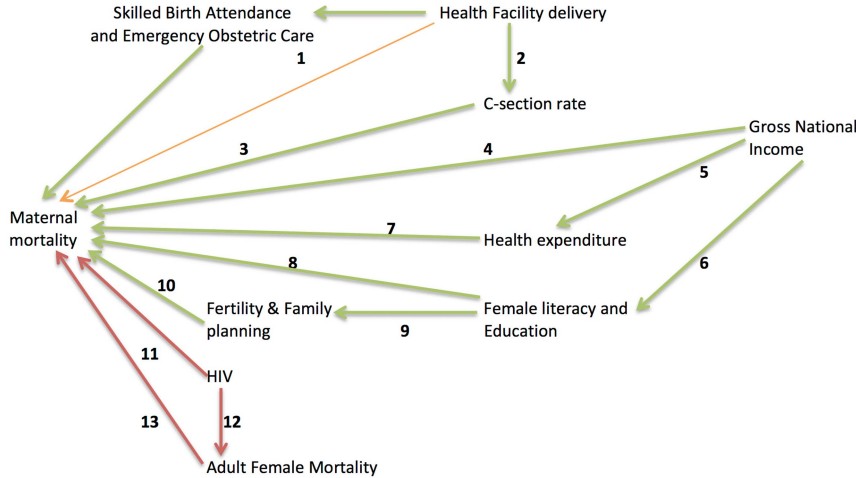

Overview of variables linked to maternal mortality in Malawi. Green arrows show beneficial effects, red arrows negative effects and orange arrows positive and negative effects

**1** Increased health facility delivery could reduce maternal mortality by improving skilled birth attendance and increasing the proportion of women covered by adequate emergency obstetric care, however, overcrowding could reduce quality of care and increase maternal case fatality rates in health facilities

**2** Caesarean-sections only occur in health facilities and are likely to increase as health facility deliveries increase. If undertaken for life-saving purposes such an increase could reduce maternal mortality

**3** As the C-section rate increases towards the WHO recommended minimum of 5% of deliveries more women who need emergency C-sections are likely to get them and therefore be more likely to survive severe complications of delivery

**4** Large increases in GNI could independently reduce maternal mortality via leading to many improvements in living conditions, as well as via increased health expenditure (**5**) and increased spending on education leading to improved female literacy and education (**6**)

**7** Increased health expenditure should improve the functioning of the health system leading to reduced maternal mortality via improved quality of antenatal, obstetric and postnatal care

**8** Increased female literacy and education should lead to reduced maternal mortality via allowing improved knowledge of maternal health problems, importance of health care and birth preparedness as well as improved family planning and reduced fertility (**9**)

**10** Increases in family planning practices will reduce fertility i.e. reduce exposure to pregnancy and child birth

**11** HIV contributes to both indirect and direct maternal mortality and via deaths from AIDS is responsible for a subset of total adult female mortality (**12**) which can be conflated with maternal mortality (**13**) if deaths co-incidental with pregnancy but not related to it are included

low it should, however, be carried out to save women's lives.[32] Health facility delivery is high in Malawi but, according to the latest national assessment, 39% of facility deliveries take place in health centres, which are poorly resourced, while 61% take place in hospitals; and the referrals of complicated cases either from home or from health centres to hospitals where blood and surgery is available is not always efficient.[33] Most C-sections in Malawi are performed by clinical officers, not by obstetricians or doctors. However, studies have revealed that clinical officers are comfortable[34] and competent[35 36] performing such operations.

### Fertility and family planning

The TFR may have an impact on maternal mortality, as women of high parity are at increased risk of potentially fatal maternal complications[37] and the lifetime risk of maternal death increases the greater the number of times a woman is exposed to pregnancy and childbirth. However, from 1982 to 2003 the TFR has changed little in comparison to the change in MMR and it has in fact even dropped slightly in the 1980s and 1990s while the MMR was increasing (table 3); therefore it is unlikely to be related to the observed trend in MMR in Malawi. The slight reduction in TFR has not impacted on the rate of population growth in Malawi, which contributes to over-burdening the health system.

Family planning methods, if accepted by a large proportion of the population, and if used continuously over prolonged periods, reduce fertility rates and consequent maternal deaths and thus contribute to reducing MMR. Family planning reduces MMR by reducing the total number of pregnancies (parity) as well as the number of unintended, unwanted and untimely pregnancies, which are often high risk.[38] Reducing unwanted pregnancies will reduce induced abortions. Abortions are estimated to be the cause of between 5% and 18% of maternal deaths according to a range of primarily hospital-based studies assessed by Bowie and Geubbels.[7] Safe abortion in Malawi is only permitted when the life of the mother is threatened; such restrictive abortion laws have not been shown to reduce maternal mortality anywhere. Success in family planning also brings about a shift in risk groups from high parity to low parity and from older age to younger age[37] but the impact of this shift on the MMR is possibly small.[39]

Although the contraceptive prevalence rate has increased over the past 20 years, this has not resulted in large changes in total fertility. Maternal mortality increased during the period of increasing contraceptive use, so they are unlikely to be related. However, it is possible that sustained gains in contraception use during the last decade, when HIV was less of a contributor to MMR (figure 1), could have reduced MMR.

**Table 3** Changes in variables hypothesised to be associated with changes in maternal mortality in Malawi

| Variable | Trend: years | | | | | |
|---|---|---|---|---|---|---|
| | 1982 | 1989 | 1997 | 2001 | 2003 | 2007 |
| MMR (approximate from figure 1) | 400 | 700 | 950 | 950 | 800 | 700 |
| Percentage deliveries by skilled attendant | | 55.5[3] | 55.6[75] | 57[45] | 54[23] | 71.4[1] |
| Percentage of deliveries by TBA | | 17.7[3] | 22.7[75] | 26.2[45] | 28.8[23] | 14.4[1] |
| Percentage of deliveries by relative/other person | | 21.8[3] | 19.0[75] | 14.2[45] | 13.0[23] | 8.7[1] |
| Percentage of deliveries alone | | 5.0[3] | 2.4[75] | 2.1[45] | 2.4[23] | 2.6[1] |
| Percentage of deliveries by C-section | | 3.4[3]* | 2.8[75]* | 3.1[45]* | 2.8[82]* | 4.6[1] |
| Total fertility rate† | 7.6[95] | 6.7[3] | 6.3[75] | 6.0[45] | 6.3[23] | 5.7[1] |
| General fertility rate‡ | 0.264[95] | 0.223[3] | 0.223[75] | 0.215[45] | 0.225[23] | 0.202[1] |
| Unmet need for FP services (%) | | 36[3] | 30[75] | 28[7] | | 26[1] |
| Contraceptive prevalence rate (%) | | 13.0[3] | 32.5[75] | 32.5[7] | 41[8] | 42.2[1] |
| GNI per capita (Atlas method, current US$) | $180[96] | $160[96] | $200[96] | $140[96] | $190[96] | $250[96] |
| GNI per capita, PPP (current international $) | $330[96] | $380[96] | $550[96] | $550[96] | $580[96] | $710[96] |
| Per capita total expenditure on health (average exchange rate US$) | | | $20[97] | $11[97] | $18[97] | $20[97]§ |
| Per capita total expenditure on health (PPP, international $) | | | $35[97] | $34[97] | $60[97] | $70[97]§ |
| Per capita Government expenditure on health (average exchange rate US$) | | | $6[97] | $7[97] | $13[97] | $15[97]§ |
| Per capita Government expenditure on health (PPP, international $) | | | $10[97] | $21[97] | $45[97] | $51[97]§ |
| External resources for health as a percentage of total expenditure on health | | | 19.5[97] | 42.4[97] | 61.6[97] | |
| Female literacy rate (ages 15 and above, %) | | 33.5[96]** | 54[96]†† | 56.5[75]‡‡ | 62.4[45]§§ | 67.6[1]¶¶ |
| Secondary school enrolment of females (gross %) | 10[96] | 11.5[96] | 20.4[96]*** | 28.8[96] | 25.5[96]††† | 26.4[96] |
| Female life expectancy at birth | 46.1[96] | 48.1[96] | 46.9[96] | 46.5[96] | 47.3[96] | 50.8[96] |
| Adult female mortality (mortality rate/1000 years exposure) | 2.6[3] | 6.5[3] | 11.3[75] | 11.6[45] | 8.7[23] | 8.4[1] |
| Percentage of adult female deaths that are maternal‡‡‡ | | 20.8[3] | 21.6[75] | 17.5[45] | 19.1[23] | 15.6[1] |
| HIV prevalence (adult population, modelled from sentinel surveillance in antenatal clinics, %) | 0[84] | 5[84] | 14[84] | 14[84] | 13[84] | 12[46] |
| Short maternal stature (% <145 cm tall) | | 2.8[3]§§§ | | 3.0[75]¶¶¶ | 3.1[45]††† | 2.4[1]**** |

*Live-births only.
†Average number of children born to a woman during her lifetime.
‡Births/number of women aged 15–44.
§Estimate for 2006.
**Estimate for 1987.
††Estimate for 1998.
‡‡Estimate for 2000. If women who can only read part of a sentence are excluded then 48.6%.
§§Estimate for 2004. If women who can only read part of a sentence are excluded then 53.8%.
¶¶Estimate for 2010. If women who can only read part of a sentence are excluded then 59.4%.
***Estimate for 1996.
†††Estimate for 2004.
‡‡‡Calculated as number of maternal deaths divided by number of adult female deaths given in separate tables in the survey reports.
§§§Estimate for 1992.
¶¶¶Estimate for 2000.
****Estimate for 2010.
GNI, gross national income; MMR, maternal mortality ratio; PPP, purchasing power parity; TBA, traditional birth attendants.

## GNI per capita

GNI per capita is consistently very low throughout the whole period and is unlikely to have contributed to significant improvements in maternal health. Given the history of maternal mortality in industrialised countries in the first half of the 20th century it is also unlikely that recent increase in GNI will have contributed to the recent reduction in MMR, unless it led to improvements in the empowerment of women to make decisions, transport and the availability of high-quality obstetric care.[6 7]

## Health expenditure

The trend in MMR may be related to the trend in health expenditure (table 3). Owing to an economic slump in 1994/1995 government health spending was reduced, and although cushioned by increases in donor funding, still dropped from 45.1 Malawi Kwacha (MKW) per capita in 1994 to 40.9 MKW per capita in 1998[40] at a time when the MMR was increasing rapidly. Also, total (government, private and individual) and government per capita health expenditure has increased in recent years, while the MMR has gone down.

## Female literacy and education

Female literacy is an indicator of formal education, and is known to be associated with lower MMR through increased knowledge of family planning and lower fertility, and increased knowledge of danger signs of pregnancy and the importance of skilled delivery.[40] At 79.7%, female literacy is higher in the Northern region of Malawi, than either the central region (64.5%) or the southern region (67.5%).[1] Despite maternity services being more sparsely distributed and under more hostile terrain the MMR is lower in northern Malawi,[23] perhaps due to the higher female literacy. For the country as a whole, there have been recent gains in adult female literacy and similar gains in the percentage of females going to secondary school (table 3), which may be related to recent declines in MMR. However, the increasing maternal mortality in the 1990s was not mirrored by declining female literacy, and the effect of literacy on MMR may follow a significant time lag—enough for other changes intermediate to reduced MMR to manifest. Such results of improved female education could include changes in family and community dynamics that give women increased agency and control over their lives.[41]

It may also be that there is a threshold level of female literacy that has to be surpassed in order for it to translate as reductions in maternal mortality via the catalysis of improved knowledge of pregnancy danger signs and the importance of skilled attendance at delivery. Above such a threshold a critical mass of knowledge may be reached within the poor communities of Malawi that contribute the most towards its high MMR.

## Female life expectancy at birth

Life expectancy at birth is an accurate proxy measure of the overall health of the population and therefore could also be an accurate predictor of maternal mortality.[13] From the available data the slight fall in female life expectancy during the 1980s and 1990s does match the rise in MMR during the same period and the rise in life expectancy in the early 2000s also matches the fall in MMR during this time, however, the trends in life expectancy are less marked (table 3). The fact that the proportion of adult female deaths that were from maternal causes appears to have remained fairly constant, varying between 21.6% and 15.6% during 1986–2010 (table 3), strengthens the case for an association between MMR and female life expectancy in Malawi. This association is plausible given that in Malawi, female life expectancy is dependent on a number of factors such as HIV and other diseases that are also linked to maternal mortality.

## Adult female mortality

Adult female mortality (AFM) follows the same trend as MMR. This is not surprising considering that MMR is a subset of AFM as determined by the sisterhood method used in the DHS and MICS. Given that AFM is around five times higher than MMR there is a large scope for the MMR to be inflated by misclassification of non-maternal adult female deaths as maternal. Sisterhood methods estimate 'pregnancy-related' deaths during pregnancy and up to 2 months postpartum, irrespective of the cause. In Bolivia, where AFM was around 11/1000 person-years between 1975 and 1988, the number of non-obstetric deaths included in MMR estimates using sisterhood methods was estimated by Stecklov[42] to be over 30%. Garenne[43] recently improved Stecklov's method by including an HIV-prevalence-adjusted estimate of the RR of non-obstetric mortality during the maternal risk period compared with the risk of mortality outside this period, and estimates the proportion of non-obstetric deaths included in the MMR for the Malawi DHS surveys to be 57% for 1992, 54% for 2000 and 62% for 2004, resulting in estimated obstetric MMRs of 299, 562 and 410, respectively. A similar upward followed by downward trend is therefore observed for this estimate of obstetric mortality as well as for pregnancy-related mortality (which was estimated to be 10–15% higher using Garenne's method than the MMR published in the DHS reports). Because the proportion of pregnancy-related deaths in HIV-positive women that are incidental deaths and the remaining proportion that are indirect obstetric deaths remains unknown,[15] these estimates of obstetric MMR could be conservative. Failure to differentiate between maternal and non-maternal deaths may lead to inaccurate conclusions about trends and impact of public health interventions, as well as inadequate future interventions.[44] Disregarding potential misclassification bias, the DHS figures show MMR to have declined by an estimated 31% (from 984 to 675 maternal deaths/100 000 live-births) and AFM to have declined by an estimated 28% (from 11.6 to 8.4 deaths/1000 women aged 15–49) between the 2004 and the

2010 surveys.[1] [45] Although the MMR decline appears slightly larger, suggesting a reduction in direct obstetric maternal deaths—as also suggested by application of Garenne's method—in addition to a reduction in HIV-related/AIDS-related maternal deaths (see below), given wide CIs, we are not able to conclude that this was the case.

## HIV, AIDS and ART

Figure 1 shows that the estimated MMR due to HIV rises dramatically during the 1990s both in absolute terms and as a proportion of the total MMR as the HIV prevalence rises (table 3). As the HIV prevalence levels off around the year 2000 and begins to slowly decline (table 3) and incidence also declines,[46] the proportion of MMR due to HIV slowly declines, declining more rapidly towards 2010 as the effect of the ART programme takes hold.[20] [21] The following paragraphs discuss the relationship between HIV and MMR in Malawi with reference to how it was calculated, biological plausibility and cause-specific mortality, regional variations and the recent effect of the ART programme.

The trend in MMR due to HIV depends on the HIV prevalence, the RR of pregnancy-related mortality in HIV-positive mothers compared to HIV-negative mothers[19] and impact of ART from 2004 onwards.[20] [47] [48] Further detail and explanation of the calculations involved are provided in web appendix 2. Although the RR of 7.74 is from a systematic review and meta-analysis of 23 studies that were mainly facility based,[19] it is corroborated by a secondary analysis of population-based longitudinal cohort data from sub-Saharan Africa that estimates it to be 8.2.[49] The fact the RR is for pregnancy-related mortality rather than maternal mortality is less of an issue given that the DHS and MICS estimates used to construct the trend line to which the RR is applied effectively measure pregnancy-related rather than maternal mortality. This also negates the issue of maternal deaths due to HIV not being formally recorded pre-2010, before the importance of HIV as an indirect cause of maternal death was recognised by the creation of International Classification of Diseases 10 (ICD10) code O98.7.[4] Recent evidence from South Africa suggests that HIV mortality is lower in pregnant women than age-matched non-pregnant women indicating that many HIV deaths may be coincidental with pregnancy.[50] Nevertheless, it remains not possible to separate out indirect obstetric deaths due to HIV and incidental HIV deaths. All are captured, however, in the national-level pregnancy-related mortality MMR trends presented in this paper. The estimated proportion of MMR attributable to HIV takes no account of the effect of the HIV epidemic on the health system of Malawi, however, which includes exacerbation of the human resource crisis,[51] and possibly also de-prioritisation of other health problems such as maternal health. Therefore it is likely to be an underestimate.

Although HIV reduces the rate of conception resulting in fewer pregnancies, it is likely to increase maternal mortality due to a combination of: increases in direct obstetric deaths (due to increases in puerperal sepsis for example); increases in indirect obstetric deaths (due to complications of HIV aggravated by pregnancy); and, decreases in the quality of care available to all mothers as a result of less trained health workers being available at health facilities (as many die from AIDS) and a more prejudicial attitude of health workers towards those who they suspect of having HIV.[52]

If HIV were responsible for an increase in maternal deaths we would expect cause-specific mortality rates to reflect this. While few studies have had the capacity to verify maternal HIV status, the proportion of deaths due to infections such as puerperal sepsis and malaria[53] may be related to HIV. Hospital-based studies have shown an increase in the proportion of maternal deaths due to puerperal sepsis from 6/78 (8%) in 1989 and 13/37 (18%) in 1990[54] to 29% in 2005[55]; both studies also finding HIV/AIDS (11% and 10% of maternal deaths in 1989 and 1990, respectively,[54] percentage not stated in 2005 study) and other infections to be important contributors. A review of 43 maternal deaths in hospitals in the central region in 2007 showed 16% were due to sepsis and attributed a further 16% to AIDS.[56] A recent study of 61 maternal deaths in a central region hospital during 2007–2011 found 12 (20%) were HIV positive, 10 of whom died of non-pregnancy-related infections including meningitis and pneumonia.[57] Such non-pregnancy-related infections would be included in the pregnancy-related mortality MMR trends presented in this paper. Another recent study of 32 maternal deaths in a tertiary hospital in Malawi in 2011 found that 13 (40%) of the women were HIV positive, 9 HIV negative and 10 had an unknown HIV status and classified 6 (19%) of the maternal deaths as due to sepsis and a further 3 (9%) due to HIV-related disease.[58] HIV infection may also predispose pregnant women to more severe malarial morbidity,[59–61] but data related to trends in malaria-related maternal complications are limited.

As noted earlier, the MMR in the northern and central regions of Malawi was significantly lower than that in the southern region of Malawi between 2000 and 2006.[23] In 2004, HIV prevalence was also significantly lower in these regions, with 10.4% (95% CI 7.8% to 13.8%), 6.6% (5.2% to 8.3%) and 19.8% (17.7% to 22.1%) in northern, central and southern regions, respectively.[45] The trend of the MMR in urban and rural areas of Malawi between the first DHS survey (1986–1992) and the second DHS survey (1994–2000) was examined by Bicego et al.[62] They found that the increases in MMR were statistically significant and concluded that they were related to increases in HIV during the same period.

There is now growing evidence from Malawi and elsewhere in sub-Saharan Africa that the population-level effect of ART is significant in reducing adult mortality rates.[47] The Malawian government Ministry of Health launched a national programme providing free access to ART in 2004. By mid-2010, 359 771 people had been

registered on ART, 345 765 in the public sector,[48] and there had been a rapid fall in mortality.[20] Therefore part of the reason for the decline in MMR in Malawi after 2003 may have been the successful scaling-up of the ART programme and a consequent reduction in direct and indirect maternal deaths related to HIV and AIDS. However, the main impact of ART is likely to have occurred later as its roll-out only became significant from 2005, with coverage reaching 25–50% by 2007–2009.[46] Although, as detailed in web appendix 2, the proportion of HIV-positive women on ART needs to increase from 35% observed in 2010[63] for there to be more of an impact. Recent reports show a large increase to 69% in the last quarter of 2012 however,[64] and the adoption of 'Option B+' of treating all HIV-positive pregnant women with antiretrovirals for life since 2011 shows great promise.[21]

### Malaria and anaemia

Malaria in pregnancy has been linked to low birth weight and adverse outcomes for the baby and severe antenatal anaemia in the mother, and antimalarial prophylaxis is recommended, especially for low-parity women.[7] Anaemia may also be unrelated to malaria, being also the result of other infections such as hookworm or HIV, or nutritional deficiencies, for example.[65] Facility-based studies in Malawi have estimated anaemia to cause 7% of 43 maternal deaths,[56] 12% of 165 maternal deaths (ref. 25, p.140), 16% of 32 maternal deaths[58] and 17% of 61 maternal deaths.[57] With regard to malaria, although studies in other malaria endemic countries have linked seasonal trends in maternal mortality to seasonal trends in malaria incidence,[66] [67] and a recent study in Kenya estimated that 9% of 249 pregnancy-related deaths were due to malaria,[68] the contribution of malaria to maternal mortality in Malawi remains unclear. Malaria control measures have increased in Malawi during the last decade of unchanged malaria transmission but have yet to reduce hospital admissions due to malaria in children.[69] [70] Current prophylaxis measures in pregnant women in Malawi may be insufficient too,[71] suggesting that the observed trend in maternal mortality in Malawi is unlikely to be due to changes in malaria or its prevention or treatment.

### Nutrition and stunting

Undernutrition during childhood and adolescence leads to stunting that puts women at risk of prolonged and obstructed labour and consequent ruptured uterus and haemorrhage, important causes of maternal mortality in Malawi.[7] The proportion of maternal deaths due to stunting remains unclear, however. Therefore, the extent to which longitudinal improvements in nutritional status could reduce maternal mortality is unknown. Given the proportion of women with short stature has changed little (table 3), stunting is unlikely to have played a major role in the observed rise and fall in MMR in Malawi.

## DISCUSSION

National estimates of maternal mortality in Malawi show a rising trend throughout the 1980s and 1990s reaching a peak in the late 1990s from which it started to decline. Maternal mortality is difficult to measure accurately, and therefore the trends observed must be interpreted with caution. However, some measure of change is necessary for monitoring progress towards achieving the MDG for maternal health. Understanding what has contributed to the rise and fall of maternal mortality is also crucial. We hope this paper goes some way towards explaining both for the case of Malawi.

### Why is the MMR falling?

A paper by Muula and Phiri[72] seeking to explain the rise in MMR between the 1992 and 2000 DHS speculated that deterioration in health services resulting from the rise in HIV during this period (and the rise in HIV itself) were possibly to blame for the apparent 80% increase in MMR during this period. The evidence we present in this paper suggests that the declining impact of HIV in Malawi may have contributed to the recent fall in maternal mortality. Muula and Phiri[72] also speculated that an increase in poverty during the 1990s could have contributed to an increase in MMR. Malawi's economy has been more successful in recent years (table 3) and poverty has perhaps decreased as a result.[73] Thus recent declines in maternal mortality could be related in part, to poverty reduction. However, the level of economic improvement required to drive a reduction in maternal mortality in Malawi remains unclear.

The functioning of the health system is key. Although the proportion of women whose delivery was attended by a skilled health-worker has only changed in recent years, the competency of the health system may have changed throughout the past three decades. An audit of maternal deaths in the southern region concluded that the quality of obstetric care went down in Malawi in the 1990s[74] and in general it is perceived that the health system deteriorated significantly during the 1990s. Although infant mortality may have decreased slightly in the 1990s, other indicators of health system performance such as vaccination rates, pneumonia treatment and stunting got worse in the 1990s.[3] [75] [76] Reasons for this decline in quality included: the human resources crisis—overseas migration of nurses and midwives peaked in the late 1990s and internal rural-to-urban migration and HIV were also responsible for declining numbers of key obstetric health workers;[51] a closure of rural facilities—some of which have since reopened following the 2004–2010 emergency human resources plan, although the Hardship Allowance for attracting staff to rural areas was not implemented;[77] declining standards in schools resulting in an insufficient number of candidates to fill nursing and medical schools; and, a lack of capable, efficient and uncorrupt leadership/strategic planning.[40] Some of these factors have improved in the last decade and could therefore be responsible for the

turnaround in MMR. Other improvements were also made since the launch of the Safe Motherhood programme in the southern region in 1997 including more health education talks and information on safe motherhood,[78] and initiatives to quicken referrals from rural health centres to hospitals.[79 80] Indicators of health system performance such as vaccination rates, pneumonia treatment and the proportion of under-5 children who are underweight or stunted have also improved in the last decade.[3 75 76]

The sector-wide approach of 2004–2010, and increased per capita health funding (table 3), has enabled increased coordination, investment, and provision of essential healthcare,[81] including some increases in nursing and other staff resulting from the Emergency Human Resources Plan.[77] However, the provision of maternity care was still less than half of what was required[81] and additional recent data suggest that obstetric care services are still lacking in Malawi, especially at the peripheral health centre level,[33 82] and also that shortages of staff and supplies are still acute.[83] The MMR would decline further if access to both basic and comprehensive emergency obstetric care could be improved further along with skilled attendance for all deliveries.[33] Clearly there is much still to do; but not just for poor rural women.

### Urban, rural, richer, poorer

The lower recent estimates from the MaiMwana and MaiKhanda population-based surveillance may also be partly due to the exclusion of urban populations from these studies.[9 10] Most health indicators in Malawi are better in urban areas,[1 23 45] but Bicego et al[62] suggest that urban maternal mortality became higher than rural maternal mortality between the 1992 and 2000 DHS, and MICS 2006 reports slightly higher MMR in urban than in rural areas (table 1), although also with widely overlapping CIs. If true, this may be partly due to higher HIV prevalence in urban areas,[62 84] and partly due to loss of the 'urban advantage' in maternal health now arising in urban populations in developing countries,[85] although a lack of access to skilled delivery in rural areas would be expected to counterbalance the lower HIV-related maternal mortality. A flattening of the socioeconomic gradient for maternal and child mortality has been seen in Malawi and other countries with high HIV prevalence, with higher mortality in low socioeconomic groups due to the lack of access to healthcare and higher mortality in high socioeconomic groups due to HIV.[86 87]

### Policy implications: reaching MDG 5 in Malawi

Assuming the MMR in 1990 was 620 (95% CI 410 to 830),[3] then the MDG 5 targeted 75% reduction for Malawi would mean an MMR of 155 (95% CI 103 to 208) by 2015.[2 88] Alternatively, using the time trend analysis from this paper (figure 1), MMR for 1990 was approximately 750 (95% CI 550 to 950) corresponding

to an MDG 5 target of 190 (95% CI 140 to 240) by 2015. Despite these uncertainties as to what the MDG 5 target should be (the countdown 2012 report sets the target for Malawi at 280,[76] basing it on the 1990 estimate of 1100, which we dispute), it is clear that meeting the target will be difficult. It is often easier to reduce mortality from a high level to a less high level than from a low level to an even lower level. Meeting MDG 5 in Malawi will also be difficult because of the rapid increase in MMR during the 1990s, possibly as a result of HIV. Both prevention and treatment of HIV are therefore also priorities in the fight against maternal mortality[89] and it is good to know that progress is now being made on both fronts.[64] If many of the extra deaths in the 1990s were due to HIV (either as incidental deaths or indirect maternal deaths complicated by HIV disease, the two of which are difficult to distinguish[90]), greater reductions in direct obstetric maternal deaths are crucial. However, the attainment of the 75% reduction target is likely to be assessed using similar sisterhood methods to the DHS and MICS, which will not adequately distinguish between pregnancy-related mortality and maternal mortality, whether due to HIV or other causes.

Translating the recent increase in institutional delivery in Malawi into increased quality of routine and emergency care during and after delivery must be prioritised as part of continued efforts towards the strengthening of the health system in Malawi.[25 33] The prevention of maternal deaths through an increased focus on family planning and liberalising safe abortion services, and improving the timeliness of referrals from homes and health centres should also be priorities.[27]

### CONCLUSION

From the best available evidence it appears that fewer mothers are dying in pregnancy, childbirth and the postpartum period in Malawi in recent years and that this is possibly the result of: increases in health facility delivery; improvements in the financing and management of the health system contributing to real gains in skilled delivery (in terms of the knowledge and skill of the increasingly available professionals involved and an enabling environment); improvements in awareness of women of the danger signs of pregnancy, catalysed through increased adult female literacy in recent years; and, recently, a reduction of HIV-related maternal mortality via a rapid roll-out of ART. Despite this it must be stressed that at around 400 maternal deaths/100 000 live-births the MMR in Malawi is still unacceptably high and much remains to be done to prevent maternal complications arising and improve the provision of obstetric care in the country. Considerable effort is required if Malawi is to achieve MDG 5 by having an MMR of 150 or less by the end of 2015 and if more mothers are going to survive in the coming years.

**Acknowledgements** The authors would like to thank Michel Garenne for his detailed and insightful review and suggestions, which improved the paper.

**Contributors** TC conceived the study, carried out the literature review, gathered the data, carried out the analyses, contributed to their interpretation, wrote the first draft of the paper, and collated inputs to all subsequent drafts. SL contributed to the analysis and interpretation and revised the paper. BN, IA, AP and CM contributed to the interpretation of the analyses and added clinical and health systems perspectives to the narrative. AP and CM also added insights from their many years working in the Malawian health system. All authors reviewed and revised several iterations of the paper, contributed intellectual content and have seen and approved the final version of the paper.

**Funding** TC and BN were funded by The Health Foundation from 2007 to 2011, and funded by the European Union, Bill and Melinda Gates Foundation, UK Government Department for International Development (DFID) and the Wellcome Trust from 2011 onwards. SL was funded by DFID and the Wellcome Trust. IA was funded by the London School of Hygiene and Tropical Medicine.

**Competing interests** None.

**Provenance and peer review** Not commissioned; externally peer reviewed.

**Data sharing statement** All of the data used in this study have been previously published.

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
