## [Reviewer comments · BMJ Open]

Some articles will have been accepted based in part or entirely on reviews undertaken for other BMJ Group journals. These will be reproduced where possible.

ARTICLE DETAILS

TITLE (PROVISIONAL)	Maternal Mortality in Malawi, 1977 to 2012
AUTHORS	Colbourn, Timothy; Lewycka, Sonia; Nambiar, Bejoy; Anwar, Iqbal; Phoya, Ann; Mhango, Chisale

VERSION 1 - REVIEW

REVIEWER	Michel Garenne Institut Pasteur, Paris
REVIEW RETURNED	16-Oct-2013

GENERAL COMMENTS	The paper estimates maternal mortality trends in Malawi from a variety of sources. Results indicate ups and down in the MMR defined as pregnancy-related deaths. The paper is well written, well argued, well documented. However, it suffers from some minor problems: 1) the proper case-definitions of pregnancy-related death, maternal death, obstetric death are used inter-changeably, creating some confusion; this is particularly important in the case of Malawi where HIV/AIDS is obviously playing a major role in trends in pregnancy-related deaths. 2) Some of the analyses and conclusions are inconsistent, and this comes from the abusive inclusion of many fortuitous causes among maternal causes. 3) Authors could better explore trends in socio-economic correlates: is they are steady, they cannot explain the ups and downs of MMR. Only HIV/AIDS seems to explain the ups and downs. These points could be corrected prior to publication. Details Abstract / Main points 1. Abstract / Results. Better to quote proper estimates from regressions. Figure 1 reads as about 250 in 1977, 700 in 1990, 1100 in 2000, 620 in 2010. Table 2 gives 748 in 1990, 970 in 2000, 484 in 2010. Readers will expect all data to be consistent, whatever the confidence interval. 2. Abstract / Results. Ups and downs in pregnancy-related deaths follow HIV mortality ups and downs. There is no evidence that these trends can be explained by anything else (no proof of deterioration followed by amelioration of health system, education, poverty, family planning, etc.) 3. Strength and limitations: the fourth point could be deleted at no cost. In first point: Maternal Mortality Ratio (not rate) Concepts 4. Concepts: Authors could make clear, from the beginning, the difference between pregnancy-related death, maternal death, and obstetric death, and stick to these definitions all over the paper. At some points the distinction is clearly made, but at others they are mixed.
---

5. The relationship with HIV could be better explained. If HIV is a risk factor for all causes-mortality as well as for pregnancy related mortality, this does not mean that pregnant women have a higher mortality than non-pregnant women when they are HIV+. Those are two different issues, which need to be distinguished. Here the emphasis is obviously on the role of HIV mortality trends on MMR trends, which is correct. Then, authors could discuss more trends in HIV prevalence among pregnant women, and ART rollout, may be in a separate Table.

Main text

6. Define abbreviations at first use (e.g. GNI, SBA)

7. Page 4, Line 53. Why not quoting the precise MMR values.

8. Page 5, Line 7 sq. The authors could explain a bit more why estimates from MMEIG and IHME differ so much, while they use the same original data.

9. Page 5, line 42. I rather see a shift from home deliveries by relative or alone to deliveries by TBA. The sentence could be rephrased.

10. Page 7, line 45 sq. It is not accurate to say that life expectancy in an accurate predictor of MMR. In fact MMR can vary very much at the same level of life expectancy, depending on the proportion of deaths of women age 15-49 due to maternal cause (which may vary from a few percent or less to 25% or more).

11. Page 7, line 55 sq. Trends in HIV determine to a large extent trends in life expectancy, in adult mortality, and in pregnancy-related mortality in Malawi.

12. Page 8, line 49 sq. I doubt that HIV can be considered as an indirect cause. It is rather a fortuitous cause, because HIV mortality during pregnancy is rather lower than outside pregnancy for women of the same age.

13. Page 9, Line 10 sq. I doubt that HIV increases mortality from malaria during pregnancy, compared with non-pregnant women of the same age. Has this been properly documented? Meningitis and pneumonia are HIV opportunistic infections. The whole paragraph could be made simpler, by focusing on the role of HIV in adult population, whether women are pregnant or not. This has a major effect on pregnancy-related mortality (which includes fortuitous HIV deaths) but not necessarily obstetric mortality.

14. Page 9, line 57 sq. Anemia is not necessarily caused by malaria. It is also a cause of maternal death in malaria free environment.

15. Discussion, Page 10, line 30sq. I doubt that the authors have clearly demonstrated a deterioration of the health system. Some critical indicators of performance (infant mortality) and functioning of the health systems (health personnel, health infrastructure, vaccination) all indicate steady improvements over the period.

16. Page 10, line 40sq. Authors should focus on ups and downs in HIV mortality to explain the ups and downs in pregnancy-related mortality. Any other cause is unlikely to explain the trend reversals, unless authors feel confident in other explanations (deficit in health personnel, etc.).

17. Page 11; line 30sq. The urban / rural divide is interesting and could be further explored: when urban mortality became higher? Relationship with HIV? Did it reverse in recent years? Are the differences significant?

18. Page 12, line 1. This sentence is typical of the confusion: if most HIV deaths during pregnancy are incidental (fortuitous), then the dynamics of the MMR will be determined by HIV treatment and prevention, and not by the dynamics in obstetric mortality.

	References: Authors could have a look at recent papers from South Africa, where trends are very similar, and for the same reason (ups and downs in HIV mortality) - Int J Womens Health. 2013 Aug 6;5:457-63. doi: 10.2147/IJWH.S45983. eCollection 2013. - PLoS One. 2013 May 13;8(5):e64414. doi: 10.1371/journal.pone.0064414. Print 2013. Tables and Figures Figure 1: is hard to read. Instead of adding several data points for period estimates, one data point at mid-period would be easier. Or present all yearly data (which would be cumbersome in this case). If the figure is printed in black and white, the data points and lines look all the same. I suggest to use various symbols. Figure 2 is useless. I suggest to delete it. Table 1: Data from DHS 1992 survey are available. So authors could compute the 95% CI for the second estimate. Table 2: Add year 1977 for current estimates, even if they do not apply for IHME and MMEIG. Table 2 and Web table 1 could be easily combined. Table 3: Is there any data on HIV mortality in Malawi? For HIV prevalence, specify 'pregnant women' if data come from surveillance sites.
--	--

REVIEWER	Sally Edmonds GP Cornwall UK up to 2012 Obstetrician in Muheza, Tanzania
REVIEW RETURNED	24-Oct-2013

GENERAL COMMENTS	A useful paper as it pulls things together nicely only it take a lot of words to do so and would have more impact if less detailed.
---

VERSION 1 – AUTHOR RESPONSE

Reviewer Name Michel Garenne

Institution and Country Institut Pasteur, Paris

Please state any competing interests or state 'None declared': None declared

The paper estimates maternal mortality trends in Malawi from a variety of sources. Results indicate ups and down in the MMR defined as pregnancy-related deaths. The paper is well written, well argued, well documented. However, it suffers from some minor problems: 1) the proper case-definitions of pregnancy-related death, maternal death, obstetric death are used inter-changeably, creating some confusion; this is particularly important in the case of Malawi where HIV/AIDS is obviously playing a major role in trends in pregnancy-related deaths. 2) Some of the analyses and conclusions are inconsistent, and this comes from the abusive inclusion of many fortuitous causes among maternal causes. 3) Authors could better explore trends in socio-economic correlates: is they are steady, they cannot explain the ups and downs of MMR. Only HIV/AIDS seems to explain the ups and downs. These points could be corrected prior to publication.

Thanks for your detailed and insightful review Michel. It is much appreciated and we have acknowledged you in the acknowledgements as believe the paper has been improved as a result of

your suggestions. Please see below for how we have addressed these three main issues (for 1 - case-definitions and HIV, see our response to points 4, 5, 12, and 13 below; for 2 – abusive inclusion of fortuitous causes, we believe given that the MMR national-level trends discussed in the paper come from the DHS and MICS surveys, which measure pregnancy-related mortality using the sisterhood method, this is less of an issue – see our responses to points 4, 12 and 18 below; for 3 – socio-economic correlates, see our responses to points 2 and 15 below)

Details

Abstract / Main points

1. Abstract / Results. Better to quote proper estimates from regressions. Figure 1 reads as about 250 in 1977, 700 in 1990, 1100 in 2000, 620 in 2010. Table 2 gives 748 in 1990, 970 in 2000, 484 in 2010. Readers will expect all data to be consistent, whatever the confidence interval.

The abstract now contains the exact figures from the best-fitting regression, which are consistent with the figures in Table 2.

2. Abstract / Results. Ups and downs in pregnancy-related deaths follow HIV mortality ups and downs. There is no evidence that these trends can be explained by anything else (no proof of deterioration followed by amelioration of health system, education, poverty, family planning, etc.) We appreciate the lack of quantitative evidence for other factors apart from HIV. This is why we have qualified our statements as being the “most plausible” considering all of the available evidence. These results and conclusions have also been influenced by the experience of the senior authors who have lived through the last 25 years of the Malawian health system.

3. Strength and limitations: the fourth point could be deleted at no cost. In first point: Maternal Mortality Ratio (not rate)

We are happy for the fourth point of the strengths and limitations to be dropped if the editor also agrees it is not necessary, and defer to the editors discretion here. However, with respect to our response to point 2 above, it may be relevant. We have corrected “rate” to “ratio”, thanks for pointing out this mistake.

Concepts

4. Concepts: Authors could make clear, from the beginning, the difference between pregnancy-related death, maternal death, and obstetric death, and stick to these definitions all over the paper. At some points the distinction is clearly made, but at others they are mixed.

Good point. We have now also included definitions of direct and indirect obstetric deaths and pregnancy-related death in the second paragraph of the introduction; and have clarified the potential conflation in surveys and with regard to HIV as follows:

“Maternal deaths can be divided into direct and indirect obstetric deaths. Direct obstetric deaths are defined as: “those resulting from obstetric complications of the pregnant state (pregnancy, labour and puerperium), from interventions, omissions, incorrect treatment, or from a chain of events resulting from any of the above” [4] and indirect obstetric deaths as “those resulting from previous existing disease or disease that developed during pregnancy and which was not due to direct obstetric causes, but which was aggravated by physiologic effects of pregnancy” [4]. Data pertaining to obstetric deaths are not always available. Deaths occurring during pregnancy, childbirth and puerperium, hereafter referred to as ‘pregnancy-related’ deaths are defined as “death occurring during pregnancy, childbirth and puerperium is the death of a woman while pregnant or within 42 days of termination of pregnancy, irrespective of the cause of death (obstetric and non obstetric)” [4]. Given a lack of adequate information on causes of death, surveys reporting maternal mortality often rely only on the timing of the death in relation to pregnancy and therefore report pregnancy-related mortality as MMR. In settings such as Malawi, where it is difficult to distinguish between HIV-disease-related indirect obstetric deaths and incidental deaths due to HIV coincident with pregnancy, this is also more

likely.”

We have also added this as the second sentence in the results: “All DHS and Multiple Indicator Cluster Survey (MICS) studies, i.e. all of the studies providing national-level estimates, use pregnancy-related deaths as case-definitions of maternal deaths to calculate MMR.”; and we have also added discussion of pregnancy-related mortality and obstetric mortality in relation to your 2011 paper in the *Studies in Family Planning* journal (we reference your figures for the Malawi DHS), after we mention Stecklov’s work in the Adult Female Mortality paragraph.

5. The relationship with HIV could be better explained. If HIV is a risk factor for all causes-mortality as well as for pregnancy related mortality, this does not mean that pregnant women have a higher mortality than non-pregnant women when they are HIV+. Those are two different issues, which need to be distinguished. Here the emphasis is obviously on the role of HIV mortality trends on MMR trends, which is correct. Then, authors could discuss more trends in HIV prevalence among pregnant women, and ART rollout, may be in a separate Table.

We have now mentioned the relationship with HIV in the introduction (see above). We are not aware of stating that “pregnant women have a higher mortality than non-pregnant women when they are HIV+” and are not concerned with non-pregnant women in this paper. We calculate the proportion of MMR likely due to HIV using the RR of pregnancy-related mortality in HIV+ mothers compared to HIV- mothers from Calvert and Ronsmans recent systematic review and meta-analysis, and the adult population HIV prevalence, again following Calvert and Ronsmans. We believe we already devote a large section of the paper to HIV prevalence (the trends of which are shown in Table 3), and ART rollout, which is also detailed in a table in Web Appendix 2.

Main text

6. Define abbreviations at first use (e.g. GNI, SBA)

GNI, SBA, DHS, MICS, UN, WHO, MKW and ICD have now all been defined at first use.

7. Page 4, Line 53. Why not quoting the precise MMR values.

Because with all the uncertainty surrounding them, the MMR values are not precise and rounded figures might be easier for readers to absorb here. We notice in your recent *Int J Womens Health* paper you also round to 300 and 670 when discussing Figure 1.

8. Page 5, Line 7 sq. The authors could explain a bit more why estimates from MMEIG and IHME differ so much, while they use the same original data.

We have added the following to the end of this paragraph:

“The MMEIG and IHME estimates differ as they are obtained from statistical models containing different covariates. The MMEIG uses Gross Domestic Product per capita (GDP), General Fertility Rate (GFR) and Skilled Birth Attendance (SBA) [1]. IHME uses analogous measures, but also uses antenatal care coverage, female education by age, ARV-adjusted HIV prevalence, neonatal death rate and malnutrition [16]. In addition, it is unclear whether the IHME estimates are based on exactly the same national-level MMR data sources as the MMEIG estimates (which use DHS and MICS), because the sources of the expanded set of sibling history data used by IHME are not disclosed in their paper or web appendix [16]. Our best-fitting estimate only uses national-level MMR data by year without using any additional covariates.”

9. Page 5, line 42. I rather see a shift from home deliveries by relative or alone to deliveries by TBA. The sentence could be rephrased.

This is what we meant. We have rephrased the sentence to now be: “Similarly, although home deliveries attended by relatives or friends have decreased, they appear to have shifted to deliveries by Traditional Birth Attendants (TBA) resulting in a similar proportion of deliveries remaining unskilled during all but the most recent years of the period in question (Table 3)”

10. Page 7, line 45 sq. It is not accurate to say that life expectancy is an accurate predictor of MMR. In fact MMR can vary very much at the same level of life expectancy, depending on the proportion of deaths of women age 15-49 due to maternal cause (which may vary from a few percent or less to 25% or more).

We base our statement that life expectancy “could be” an accurate predictor of maternal mortality on Ralph Hakkert’s model of maternal mortality, which includes life expectancy as a covariate, as referenced in the sentence: “Life expectancy at birth is an accurate proxy measure of the overall health of the population and therefore could also be an accurate predictor of maternal mortality [2]”. As explained in the methods, the potential drivers of maternal mortality that we chose to investigate came, among other sources, from those of modelling studies by others such as Ralph Hakkert. We take your point. However, from the DHS and MICS data it seems the proportion of deaths that are due to a maternal cause has remained fairly constant in Malawi, between 15.6% and 21.6% during the period in question. We have added this to Table 3 and amended the last sentence of the paragraph to say: “The fact that the proportion of adult female deaths that were from maternal causes appears to have remained fairly constant, varying between 15.6% and 21.6% during 1986 to 2010 (Table 3), strengthens the case for an association between MMR and female life expectancy in Malawi. This association is plausible given that in Malawi, female life expectancy is dependent on a number of factors such as HIV and other diseases that are also linked to maternal mortality.

11. Page 7, line 55 sq. Trends in HIV determine to a large extent trends in life expectancy, in adult mortality, and in pregnancy-related mortality in Malawi.

We agree, however are not sure whether you are requesting a change in wording here? In this sentence we point out that Adult Female Mortality (AFM) follows the same trend as MMR in Malawi, which is what the DHS data shows.

12. Page 8, line 49 sq. I doubt that HIV can be considered as an indirect cause. It is rather a fortuitous cause, because HIV mortality during pregnancy is rather lower than outside pregnancy for women of the same age.

We believe you are referring to the results of your recently published papers, cited below. Thanks for drawing our attention to this. We were merely using the ICD10 definition of indirect maternal death due to HIV (code O98.7). We have now gone on to discuss this in the context of your findings, referencing your work. We have added the following:

“Recent evidence from South Africa suggests HIV mortality is lower in pregnant women than age-matched non-pregnant women indicating that many HIV deaths may be co-incident with pregnancy [49]. Nevertheless, it remains not possible to separate out indirect obstetric deaths due to HIV and incidental HIV deaths. All are captured, however, in the national-level pregnancy-related mortality MMR trends presented in this paper.”

With regard to indirect obstetric deaths due to HIV, we think these are still possible even if HIV mortality is lower in pregnancy than non-pregnancy, i.e. it’s possible that without indirect obstetric deaths due to HIV, the risk of death in HIV+ women who are pregnant would be even lower than in age-matched HIV+ women who are not pregnant.

13. Page 9, Line 10 sq. I doubt that HIV increases mortality from malaria during pregnancy, compared with non-pregnant women of the same age. Has this been properly documented?

An expert review (new reference [51] Gonzalez et al 2012) suggests an increase in malaria mortality associated with HIV in pregnancy. We have also now found and referenced studies that show an increase of malaria in pregnancy in the HIV infected (see new refs [57-59]).

Meningitis and pneumonia are HIV opportunistic infections.

We already characterise meningitis and HIV as “non-pregnancy-related infections”. We have added a sentence to make it clear that even though this is the case they would be included in the deaths that make up the MMR trends discussed in the paper: “Such non-pregnancy-related infections would be

included in the pregnancy-related mortality MMR trends presented in this paper.”

The whole paragraph could be made simpler, by focusing on the role of HIV in adult population, whether women are pregnant or not. This has a major effect on pregnancy-related mortality (which includes fortuitous HIV deaths) but not necessarily obstetric mortality.

Instead of comparing pregnant with non-pregnant women, which is complicated by the selection effect of healthier women becoming pregnant in the first place, we would rather stick to the discussion of morbidity and mortality related to infections in pregnancy that may also be exacerbated by HIV. The subject of our paper remains pregnant women, rather than the whole adult population. We hope the changes and clarifications we've added in responding to your earlier points about HIV (see above) mean that this paragraph makes better sense.

14. Page 9, line 57 sq. Anemia is not necessarily caused by malaria. It is also a cause of maternal death in malaria free environment.

Good point, we have added a sentence: “Anaemia may also be unrelated to malaria, being also the result of other infections such as hookworm or HIV, or nutritional deficiencies, for example [63].”; and have also amended the sentence two sentences later to start as : “With regard to Malaria...” in order to make this clear.

15. Discussion, Page 10, line 30sq. I doubt that the authors have clearly demonstrated a deterioration of the health system. Some critical indicators of performance (infant mortality) and functioning of the health systems (health personnel, health infrastructure, vaccination) all indicate steady improvements over the period.

In the discussion we argue (with reference to key studies) for a deterioration in the health system, especially in relation to obstetric care in the 1990s and then an improvement in the last decade. The key indicators you mention did not really show steady improvements in the 1990s in our opinion – infant mortality may have went down slightly, but vaccination coverage also went down slightly as did pneumonia treatment, and stunting also perhaps increased. There were only significant improvements in these indicators in the last decade or so, which matches our thinking in the discussion. We have added two sentences to the third paragraph of the discussion: “Although infant mortality may have decreased slightly in the 1990s, other indicators of health system performance such as vaccination rates, pneumonia treatment and stunting got worse in the 1990s [3 74 75].”

and at the end of the paragraph:

“Indicators of health system performance such as vaccination rates, pneumonia treatment and the proportion of under-5 children who are underweight or stunted have also improved in the last decade [3 74 75].”

16. Page 10, line 40sq. Authors should focus on ups and downs in HIV mortality to explain the ups and downs in pregnancy-related mortality. Any other cause is unlikely to explain the trend reversals, unless authors feel confident in other explanations (deficit in health personnel, etc.).

We have focused on HIV a lot in the paper, but also consider other important factors. We feel as confident as we have stated in the other potential explanations (i.e. we have qualified our statements and presented our best summary of the available evidence, and our judgement is also based on our experience). Please note we have also linked HIV to the deterioration of the health system in the 1990s.

17. Page 11; line 30sq. The urban / rural divide is interesting and could be further explored: when urban mortality became higher? Relationship with HIV? Did it reverse in recent years? Are the differences significant?

It's possible that urban mortality became higher between the 1992 DHS (1986-1992) and the 2000 DHS (1994-2000), although the confidence intervals overlap a lot. As we state in the discussion it could be due to HIV, but also other factors. We have revised the second and third sentences of this paragraph to reflect this:

“Most health indicators in Malawi are better in urban areas [1 23 44], but Bicego et al suggest that urban maternal mortality became higher than rural maternal mortality between the 1992 and 2000 DHS [60], and MICS 2006 reports slightly higher MMR in urban than in rural areas (Table 1), although also with widely overlapping confidence intervals. If true, this may be partly due to higher HIV prevalence in urban areas [60 80], and partly due to the loss of ‘urban advantage’ in maternal health now arising in urban populations in developing countries [81], although a lack of access to skilled delivery in rural areas would be expected to counterbalance the lower HIV-related maternal mortality.” Unfortunately we don’t have any data on the urban/rural split for the last 5 years when ART really got going (and even if we did, it would also have wide overlapping confidence intervals), so we can’t tell whether the urban/rural differences may have reversed in recent years.

18. Page 12, line 1. This sentence is typical of the confusion: if most HIV deaths during pregnancy are incidental (fortuitous), then the dynamics of the MMR will be determined by HIV treatment and prevention, and not by the dynamics in obstetric mortality.

Are you saying here that HIV will be the driver because MMR (as measured for the MDG 5 target) is really pregnancy-related mortality? We were trying to indicate that if incidental HIV deaths are not maternal deaths – and are therefore not counted in MMR (although this is not currently the case in i.e. the DHS and MICS measure pregnancy-related mortality, which is therefore the trend discussed in the paper, as we now make clear in several added sentences throughout the paper) - then reducing them won’t impact on MMR made up of ‘true maternal deaths’ i.e. direct obstetric maternal deaths, which may not have come down like the MMR appears to have. We agree it sounds confused and have changed the sentence to the following:

“If many of the extra deaths in the 1990s were incidental HIV deaths rather than indirect maternal deaths complicated by HIV-disease, greater reductions in direct obstetric maternal deaths are crucial. However, the target for attaining the 75% reduction is likely to be measured using similar sisterhood methods to the DHS and MICS, which will not adequately distinguish between pregnancy-related mortality and maternal mortality, whether HIV-related or otherwise.”

References:

Authors could have a look at recent papers from South Africa, where trends are very similar, and for the same reason (ups and downs in HIV mortality)

- Int J Womens Health. 2013 Aug 6;5:457-63. doi: 10.2147/IJWH.S45983. eCollection 2013.

- PLoS One. 2013 May 13;8(5):e64414. doi: 10.1371/journal.pone.0064414. Print 2013.

Thanks for these. We have referenced them in our discussions of HIV and MMR. They are references [49] and [89]. Incidentally, when reading through the PLoS One paper, I noticed some mistakes e.g. 95%IC instead of 95%CI, and the author contributions seem to be for a different paper (e.g. it talks about cell lines, and who are CS and AL?)

Tables and Figures

Figure 1: is hard to read. Instead of adding several data points for period estimates, one data point at mid-period would be easier. Or present all yearly data (which would be cumbersome in this case). If the figure is printed in black and white, the data points and lines look all the same. I suggest to use various symbols.

We appreciate the complexity of Figure 1 may make it hard to digest. However, we included yearly data points for period estimates so that the reader appreciates that most survey estimates apply to a number of years (something we believe is under-appreciated with respect to estimates of maternal mortality); and also because, as explained in Web Appendix 1, that was how the regression of year against MMR estimates, used to calculate the best-fitting line, was plotted.

Thanks for pointing out problems with black and white printing. We have amended the lines on the figures so that they are different patterns of dots and dashes (as well as different colours – Editor, we would like the Figure 1 to be published in colour please); and have also changed the symbol shapes to distinguish the different regions and the MMR due to HIV.

Figure 2 is useless. I suggest to delete it.

We respect your opinion Michel. However, we believe it is a useful conceptual overview to help the reader digest the substantial material in the text. We are happy to discuss omitting Figure 2 if the editor also believes it is superfluous.

Table 1: Data from DHS 1992 survey are available. So authors could compute the 95% CI for the second estimate.

The main estimate for the DHS 1992 survey is included as line 6 of Table 1 and has 95%CI. The second line that the reviewer is referring to is for a re-analysis of the data pertaining to an earlier period (1977-1983) – we were unable to obtain the data used in this re-analysis.

Table 2: Add year 1977 for current estimates, even if they do not apply for IHME and MMEIG. Table 2 and Web table 1 could be easily combined.

The 1977 best-fit estimate has now been added to Table 2.

We believe Web Table 1 is better where it is in Web Appendix 1, as the additional row requires the lengthy explanation contained within Web Appendix 1.

Table 3: Is there any data on HIV mortality in Malawi? For HIV prevalence, specify 'pregnant women' if data come from surveillance sites.

HIV mortality data is available from UNAIDS as number of AIDS deaths per year and the trend broadly reflects that observed in Adult Female Mortality, which is already in Table 3 and discussed in relation to AIDS mortality in the paper.

The HIV prevalence data in Table 3 are national estimates for the whole adult population modelled from sentinel antenatal surveillance sites (pregnant women). We have added this to the table to make it clearer.

Reviewer Name Sally Edmonds

Institution and Country GP Cornwall UK

up to 2012 Obstetrician in Muheza, Tanzania

Please state any competing interests or state 'None declared': None declared

A useful paper as it pulls things together nicely only it take a lot of words to do so and would have more impact if less detailed.

Thanks for your review. We have endeavoured to keep the paper as brief as possible. However, we believe the current length is required in order to explain things adequately. Indeed as you'll appreciate from the first reviewers comments it could be argued that more detail was required.

VERSION 2 – REVIEW

REVIEWER	Michel Garenne Institut Pasteur, Paris
REVIEW RETURNED	07-Nov-2013

GENERAL COMMENTS	The paper is now ready for publication. The authors have properly addressed all the points raised in the first review.
---